# Evaluating the potential impact of rubella-containing vaccine introduction on congenital rubella syndrome in Afghanistan, Dem. Republic of Congo, Ethiopia, Nigeria, and Pakistan: A mathematical modeling study

**Sebastian A. Rodriguez-Cartes**[1], **Yiwei Zhang**[2], **Maria E. Mayorga**[1,2]*, **Julie L. Swann**[1,2], **Benjamin T. Allaire**[3]

1 Edward P. Fitts Department of Industrial & Systems Engineering, North Carolina State University, Raleigh, North Carolina, United States of America, 2 Operations Research Program, North Carolina State University, Raleigh, North Carolina, United States of America, 3 RTI International, Research Triangle Park, North Carolina, United States of America

* memayorga@ncsu.edu

**Data Availability Statement:** All data are publicly available; chosen data points as well as parameters

## Abstract

We assessed the potential impact of introducing rubella-containing vaccine (RCV) on congenital rubella syndrome (CRS) incidence in Afghanistan (AFG), Democratic Republic of Congo (COD), Ethiopia (ETH), Nigeria (NGA), and Pakistan (PAK). We simulated several RCV introduction scenarios over 30 years using a validated mathematical model. Our findings indicate that RCV introduction could avert between 86,000 and 535,000 CRS births, preventing 2.5 to 15.8 million disability-adjusted life years. AFG and PAK could reduce about 90% of CRS births by introducing RCV with current measles routine coverage and executing supplemental immunization activities (SIAs). However, COD, NGA, and ETH must increase their current routine vaccination coverage to reduce CRS incidence significantly. This study showcases the potential benefits of RCV introduction and reinforces the need for global action to strengthen immunization programs.

## 1. Introduction

Worldwide, an estimated 100,000 infants are born yearly with congenital rubella syndrome (CRS) [1]. Although rubella produces mild symptoms in most of the population and antibodies acquired through natural infection generate life-long immunity, virus circulation must be interrupted through vaccination to eliminate rubella and prevent CRS births. The introduction of rubella-containing vaccines (RCV) depends significantly on the countries' commitment to achieve rubella elimination and available financial support from global partners, with 21 countries yet to introduce RCV by 2020 [2]. After the introduction, maintaining sufficient vaccination coverage presents additional challenges. For example, during the COVID-19 pandemic, countries reduced the utilization of antenatal care and immunization, and lockdowns

used are in the manuscript and/or supporting information files.

**Funding:** This work was supported in part by the Centers for Disease Control and Prevention Foundation(contract number HHSD2002013M53964B under Task Order 75D30120F09850 to SAR, YZ, MEM, JLS, BTA). The funders supported the implementation of the mathematical model for internal use. The authors extended the model for this study and the funders had no role in study design, data collection and analysis, decision to publish, or preparation of this manuscript.

**Competing interests:** The authors have declared that no competing interests exist.

negatively impacted routine vaccination coverage of children, setting back gains in routine vaccination coverage experienced in the late 2010s [3–5].

The Centers for Disease Control and Prevention's Global Immunization Division prioritizes countries with the highest burden of vaccine-preventable diseases and classifies them as Tier 1 countries [6]. As of 2021, the Tier 1 countries that have not introduced RCV in their Expanded Program on Immunization (EPI) schedules are Afghanistan (AFG), the Democratic Republic of the Congo (COD), Ethiopia (ETH), and Nigeria (NGA). Pakistan (PAK) introduced RCV in 2022. The high burden of rubella infection in these countries has been widely studied, which supports the need for RCV introduction [7–14]. Moreover, all these countries have included MCV1 and MCV2 (the first and second doses of measles-containing vaccine) in their routine vaccination schedules, except for the Democratic Republic of Congo, which has not included MCV2. The availability of combination vaccines such as measles-rubella (MR) and measles-mumps-rubella (MMR) facilitates the introduction of rubella-containing vaccines on immunization schedules.

Several mathematical models have been developed to model the dynamics of rubella and estimate the potential impacts of vaccination on the number of CRS births. Catalytic models have been developed to estimate the number of CRS births in developing countries [15]. A structured population model has been presented to analyze the impact of demography on immunization scenarios [16]. Additionally, studies have been conducted to combine mathematical models with serological data to estimate the burden of CRS in several countries [17,18]. These studies serve as valuable information, considering the need for more reliable reporting data in developing countries. Studies have been conducted to support the investment in measles and rubella elimination or control by doing a cost-benefit analysis [19].

Moreover, other studies were developed to evaluate immunization scenarios by estimating the burden of congenital rubella syndrome (CRS) incidence in specific countries, e.g., Madagascar, Vietnam, South Africa, and Dem. Republic of Congo [20–23]. Nonetheless, additional efforts are required to evaluate the potential impact of RCV introduction in Tier 1 countries, with prior efforts providing a validated modeling framework.

This study aims to investigate several strategies for RCV introduction in the selected countries. We demonstrate a model and estimate the potential impact of RCV introduction on CRS incidence under several types and levels of routine immunization and supplemental immunization activities (SIAs). Additionally, we conducted a sensitivity analysis to quantify the effect of the force of infection estimates in our findings. Results serve as guidance for policymakers aiming to incorporate RCV into their immunization schedules, considering the current routine coverage levels in each country, and to better inform decisions regarding resource allocation towards the eradication of CRS.

## 2. Materials and methods

### 2.1 Compartmental model

We built a country-level, deterministic compartmental model stratified by age and sex, as described by [17], to characterize the dynamics of rubella infections and CRS births in our selected group of countries: AFG, COD, ETH, NGA, and PAK. The model considers the epidemiological states in each country, defined as maternally immune (*M*), susceptible (*S*), exposed (*E*), infected (*I*), and recovered (*R*). The population in the *M* state corresponds to newborns that acquired immunity passively through their mother, lasting six months in the model. Later, individuals either become susceptible to rubella (*S*) or immune due to vaccination (*R*). The *R* state corresponds to the individuals who acquired immunity by natural infection or vaccination. The *E* state represents individuals exposed to rubella (e.g., through contact with someone

infectious) who have been effectively infected but are not yet infectious. The *I* state represents the population that is infectious to others. Infections are driven by a time-dependent force of infection, which is estimated for both young ($<\,= 13$ years old) and old ($> 13$ years old) populations. The size of age groups comprises one year, and transitions between age groups happen at the beginning of each year. We assumed that routine immunization was done annually over each cohort, with two routine RCV doses applied after children lose their maternal immunity and when transitioning to the next age group, respectively. We also assumed that SIA campaigns are conducted at the beginning of each year. These assumptions ensure the proper immunization of cohorts. Additionally, we incorporated the possibility of imported cases by introducing 10 cases per year. The general equations and parameters that govern the transition between states and age groups are described in S1 Appendix.

We used demographic data from a variety of sources. We gathered 2020 crude birth rates from the World Bank [24] and age-specific death rates from WHO 2019 life tables [25]. Additionally, we utilized the birth estimates from the UN's 2019 world population prospects [26].

## 2.2 RCV introduction

We quantified the potential effect of RCV introduction in AFG, COD, ETH, NGA, and PAK under different immunization scenarios considering routine and SIA vaccinations. The introduction year was assumed to be 2021.

For coverage and schedule, we use assumptions based on existing country schedules. First, we considered that the introduction of RCV on each EPI schedule can be implemented through measles-containing combination vaccines. For countries that have implemented two doses of measles-containing vaccines (MCV1 and MCV2), we assumed the vaccination coverage of the first and second routine RCV doses (RCV1 and RCV2, respectively) were capped by the WHO-UNICEF 2019 coverage estimates for MCV1 and MCV2 in each country [27]. For those countries where no MCV2 has been introduced yet (COD), we used a conservative estimate of 30% for the MCV2 routine coverage based on other countries within the same region that reported similar MCV1 coverage (WHO-UNICEF). Additionally, we assumed that during the first year of introduction, the RCV coverage would correspond to 50% of the MCV immunization coverage. For the remainder of the planning horizon, coverage was kept constant at 100% of the current MCV coverage.

For SIAs, we assumed that they all achieve the WHO's recommended minimum coverage level of 80% [1]. We assumed that SIAs often reach those vaccinated through routine immunization [28]. Thus, the effective vaccination rate for age groups targeted by both routine immunization and SIAs was calculated by multiplying the maximum between both coverages with the vaccine efficacy. We also assumed that no RCV vaccination campaign had been developed in previous years and that the population had reached an endemic equilibrium in each country prior to RCV introduction. The latter refers to the proportion of the population in each state when the compartmental model reaches the asymptotically stable equilibrium with the disease not being eradicated from the population, given an underlying average force of infection.

**2.2.1 RCV introduction scenarios.** We defined several introduction scenarios to understand the potential impact of different vaccination strategies on CRS incidence. We built these scenarios considering the activities recommended in the WHO guidelines for introducing RCV [1,29], which include vaccination campaigns targeting a comprehensive portion of children and using at least one RCV routine dose.

We considered immunization through three different sources: routine vaccination, catch-up SIA, and follow-up SIAs. We assumed that routine immunization consists of two independent RCV doses. We define a catch-up SIA as corresponding to an extensive vaccination

campaign conducted during the first year of introduction targeting 0–14 year olds, and follow-up SIAs refer to vaccination campaigns carried out after introducing the new antigen targeting 0–4 year olds. Note that the 0–1 year old age group is considered within these targets, given that the lower bound target of SIAs is children older than nine months old. We extended the RCV introduction scenarios developed in [22] and defined five different RCV introduction scenarios based on the combinations of those vaccination policies; see Table 1. We selected these scenarios as they represent different efforts toward eliminating CRS births. For scenarios where follow-up SIAs are conducted, we determined their frequency based on the targeted age groups, corresponding to a 5-year cohort. We also defined a zero-vaccination scenario, which serves as a baseline for comparison.

**2.2.2 Outcomes.** The primary model outcome is CRS incidence per 100,000 live births over 30 years. We also estimated the number of CRS births by considering the three variant levels of projected births (low (L), medium (M), and high (H)) from [26]. Furthermore, we quantified Disability Adjusted Life-Years (DALYs) incurred by multiplying the number of CRS births averted and the undiscounted DALYs lost per CRS case to estimate the disease burden caused by rubella. Specifically, we used DALYs estimated by [30] using the 2010 Global Burden of Diseases (GBD) disability weights for low-income and lower-middle-income countries, corresponding to 29 and 30 DALYs per CRS case, respectively. Each country was associated with the corresponding parameter based on their latest World Bank income level [31]. In addition, we calculated the CRS births averted and DALYs averted by comparing the CRS births and DALYs under S1-S5 with the baseline scenario S0. Because outcome measures are unavailable for the countries of interest, we validated the model by comparing results with previous studies (see S1 Appendix).

### 2.3 Experiments

We evaluated the potential impact of the proposed scenarios on the CRS incidence and quantified the CRS births and DALYs averted. We also analyzed the importance of including RCV2 in the EPI schedule. We examined the expected CRS births of administering one or two RCV doses while quantifying the total input doses for each scenario.

Then, to evaluate decision-making from a policymaker's perspective, we used Scenario S4 in the following experiments due to its similarity to actual vaccine introduction following WHO guidelines. We analyzed the reduction in CRS incidence when the RCV1 introduction coverage recommended by WHO is achieved. We compared these results with eight levels of coverage ranging from 60% to 95% and discussed the implications of increasing routine vaccination coverage. For simplicity, we assumed that RCV2 remained unchanged at each country's predicted level.

We also conducted a sensitivity analysis on the average force of infection estimated for each country. We sampled 100 transmission parameters from the 95% confidence intervals of the average force of infection estimates [17,18] using bootstrapping and projected outcomes to create quantile intervals for the CRS incidence (see Table B in S1 Appendix). To maintain feasibility, we discarded those parameters where the old population's force of infection was higher than for the young one. Additional analysis of the potential impact of the frequency of follow-up SIAs in high-transmission countries is presented in S2 Appendix.

## 3. Results

### 3.1 Evaluating RCV introduction policies

Fig 1 shows the estimated CRS incidence for 30 years after the introduction of RCV under the proposed scenarios. For scenario S0, the CRS incidence of each country remained

**Table 1. Definition of RCV introduction scenarios, with each one representing a combination of routine vaccination in the vaccination schedule as well as additional catch-up and follow-up SIAs.**

| Scenario | Routine vaccination | Catch-up SIA | Follow-up SIAs |
|---|---|---|---|
| **Scenario 0 (S0)** - No introduction of RCV in the vaccination schedule | No | No | No |
| **Scenario 1 (S1)** - Routine vaccination only. | Yes | No | No |
| **Scenario 2 (S2)** - Routine vaccination with a catch-up SIA. | Yes | Yes | No |
| **Scenario 3 (S3)** - Routine vaccination with multiple follow-up SIAs. | Yes | No | Yes, SIAs every 5 years over 30 years (5 SIAs). |
| **Scenario 4 (S4)** - Routine vaccination with a catch-up SIA and a single follow-up SIAs. | Yes | Yes | Yes, 1 SIA after 5 years of introduction. |
| **Scenario 5 (S5)** - Routine vaccination with a catch-up SIA and multiple follow-up SIAs. | Yes | Yes | Yes, SIAs every 5 years over 30 years (5 SIAs). |

approximately constant, considering that the pre-vaccination equilibrium had been reached. This serves as a line of comparison to quantify the effects of RCV introduction. Without any vaccination in place (S0 scenario), we estimate the expected CRS births in steady state to range from 886 (AFG) to 10,047 (NGA) each year with associated DALYs of 773,122 and 8,997,099 respectively for the 30 years. Scenario S1 shows that in countries with lower routine vaccination coverage (NGA and ETH) and no SIAs, CRS incidence rises to similar pre-vaccination levels after five years of introduction. The effects of low levels of routine vaccination coverage can be seen in scenario S1, where outbreaks occur every seven years after the introduction in NGA and after 15 years of introduction in ETH.

Similar consequences can be seen in scenario S2. Other countries (AFG and COD) present increases in CRS incidence by the end of the study period. PAK achieves a significant decrease in CRS incidence. On the other hand, scenarios S2, S3, and S4 present similar outcomes for NGA and ETH, where the initial catch-up and follow-up SIAs can significantly decrease CRS incidence, yet infections rise again after ten years. S5 can decrease CRS births close to zero for all countries, which is expected considering that SIAs are conducted during the entire time horizon. Despite the continuous aid in immunization, NGA presents an increase in CRS incidence because of low routine coverage. It is important to note that Scenario S4 achieves similar results with a single follow-up SIA for AFG.

The results on CRS births averted, and DALYs averted are shown in Table 2. At least 86 thousand CRS births (77–95 for the low and high variants of UN's population estimates) can be averted when comparing introducing routine vaccination (S1) against zero vaccination (S0). Introducing RCV at low coverages produces more CRS births than the no-vaccination scenario in NGA and ETH. The estimate increases to 535 thousand CRS births averted (462–610) when comparing the RCV introduction supported with long-term SIAs (S5) against zero vaccination (S0). This translates to averting between 2.5 (2.2–2.8) and 15.8 (13.7–18.0) million DALYs across the five countries.

Results show that S2, S4, and S5 performed similarly in reducing the number of CRS births and DALYs for AFG, COD, and PAK at all the variant levels. For ETH and NGA, S5 is the best at averting the number of CRS births and DALYs at all three variant levels.

## 3.2 Evaluating the potential impact of RCV2

Although the WHO recommends a single RCV dose to be part of the immunization schedule as the vaccine efficacy is greater than 95% [1], we analyzed the potential impact in CRS births averted of including a second dose of RCV in each country's EPI schedule. Fig 2 presents the potential impact of incorporating RCV2 in the routine immunization schedule compared to

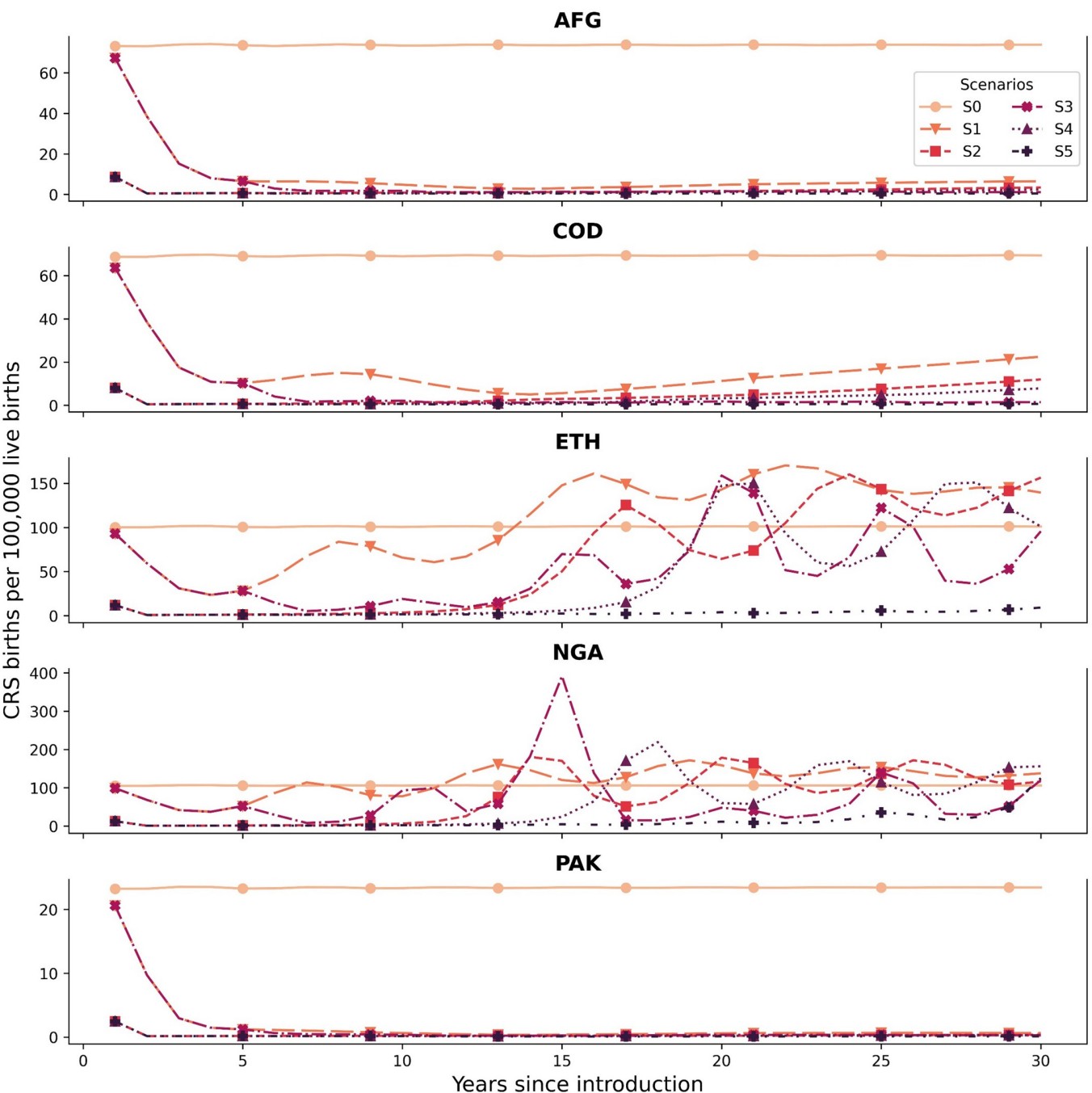

**Fig 1. CRS incidence under different scenarios for each country.** An illustration of the distribution of rubella cases across age groups for each scenario is presented in S3 Appendix.

the same scenario with only a single routine dose. As RCV2 targets children who got a first dose, countries with relatively high RCV1 coverage, such as AFG and PAK, obtain a low marginal benefit (less than 1% increase in CRS births averted) when looking at scenarios accompanied with SIAs (S2, S3, S4, and S5). In COD, a second routine dose significantly impacts scenario S1, with over 1700 more cases averted. On the other hand, RCV2 has a more significant impact on ETH and NGA with higher marginal benefits than just using a single routine

**Table 2. Comparison of CRS births averted and DALYs averted of each country under scenarios S1-S5, each with two routine doses.** See Table A in S3 Appendix for CRS births and DALYs estimated for scenario S0.

| Scenario | Projection | AFG | | COD | | ETH | | NGA | | PAK | |
|---|---|---|---|---|---|---|---|---|---|---|---|
| | | CRS births averted | DALYs averted | CRS births averted | DALYs averted | CRS births averted | DALYs averted | CRS births averted | DALYs averted | CRS births averted | DALYs averted |
| S1 | L | 19,216 (88%) | 557,256 | 65,123 (78%) | 1,888,559 | -6,474 (-7%) | -187,735 | -32,727 (-12%) | -981,815 | 31,798 (92%) | 953,932 |
| | M | 23,515 (88%) | 681,935 | 74,464 (78%) | 2,159,461 | -10,579 (-9%) | -306,803 | -40,424 (-13%) | -1,212,734 | 39,161 (93%) | 1,174,824 |
| | H | 27,942 (89%) | 810,328 | 84,062 (78%) | 2,437,802 | -14,960 (-11%) | -433,830 | -48,429 (-14%) | -1,452,867 | 46,795 (93%) | 1,403,853 |
| S2 | L | 21,437 (98%) | 621,679 | 78,071 (94%) | 2,264,067 | 39,518 (40%) | 1,146,009 | 69,115 (26%) | 2,073,440 | 34,087 (99%) | 1,022,608 |
| | M | 26,028 (98%) | 754,803 | 88,955 (94%) | 2,579,709 | 43,651 (37%) | 1,265,878 | 72,606 (24%) | 2,178,176 | 41,716 (99%) | 1,251,475 |
| | H | 30,752 (98%) | 891,816 | 100,143 (93%) | 2,904,159 | 47,607 (34%) | 1,380,594 | 75,864 (22%) | 2,275,913 | 49,620 (99%) | 1,488,594 |
| S3 | L | 20,053 (91%) | 581,533 | 76,609 (92%) | 2,221,655 | 48,300 (49%) | 1,400,704 | 87,240 (33%) | 2,617,194 | 32,152 (93%) | 964,555 |
| | M | 24,566 (92%) | 712,416 | 87,925 (92%) | 2,549,837 | 56,662 (48%) | 1,643,205 | 98,328 (33%) | 2,949,841 | 39,604 (94%) | 1,188,108 |
| | H | 29,216 (93%) | 847,269 | 99,583 (93%) | 2,887,901 | 65,158 (47%) | 1,889,583 | 109,687 (32%) | 3,290,614 | 47,331 (94%) | 1,419,919 |
| S4 | L | 21,559 (98%) | 625,221 | 79,960 (96%) | 2,318,854 | 54,904 (56%) | 1,592,219 | 99,782 (38%) | 2,993,454 | 34,145 (99%) | 1,024,352 |
| | M | 26,184 (98%) | 759,340 | 91,192 (96%) | 2,644,582 | 62,720 (53%) | 1,818,891 | 107,977 (36%) | 3,239,295 | 41,790 (99%) | 1,253,700 |
| | H | 30,945 (98%) | 897,397 | 102,744 (96%) | 2,979,580 | 70,483 (51%) | 2,044,005 | 116,047 (34%) | 3,481,422 | 49,711 (99%) | 1,491,329 |
| S5 | L | 21,708 (99%) | 629,533 | 82,375 (99%) | 2,388,873 | 95,102 (97%) | 2,757,945 | 228,816 (87%) | 6,864,489 | 34,221 (99%) | 1,026,636 |
| | M | 26,379 (99%) | 764,979 | 94,088 (99%) | 2,728,558 | 114,139 (97%) | 3,310,036 | 258,884 (86%) | 7,766,509 | 41,889 (99%) | 1,256,674 |
| | H | 31,188 (99%) | 904,442 | 106,147 (99%) | 3,078,253 | 133,772 (97%) | 3,879,381 | 289,525 (86%) | 8,685,753 | 49,835 (99%) | 1,495,038 |

dose. This benefit translates to averting approximately 1,000 and 2,000 more CRS births in ETH and between 1,000 and 3,000 in NGA. Results highlight a benefit in incorporating RCV2, especially if it can be done through combination vaccines at the current MCV2 levels.

### 3.3. Effect of vaccination coverage on CRS incidence

Fig 3 presents the number of CRS births over time (per 100,000 live births) for different RCV1 introduction levels for scenario S4. We highlighted the minimum immunization RCV1 coverage recommended by the WHO of 80% [1] and used it as a point of comparison. Note that the magnitude of the variation in CRS incidence due to increased routine coverage differs across countries. AFG, COD, and PAK see no significant increase in CRS incidence for coverages over 80%. Considering the underlying assumptions regarding the force of infection dynamics, countries with high infectious sero-profiles in both young and old populations, such as NGA and ETH, will see late outbreaks when routine coverage is below 85%. Moreover, these countries show consistently high surges of CRS incidence for immunization coverages below 80%. The timing of the peaks of high incidence varies, with lower coverage levels producing earlier outbreaks. A lower incidence is expected at 90% RCV1 coverage.

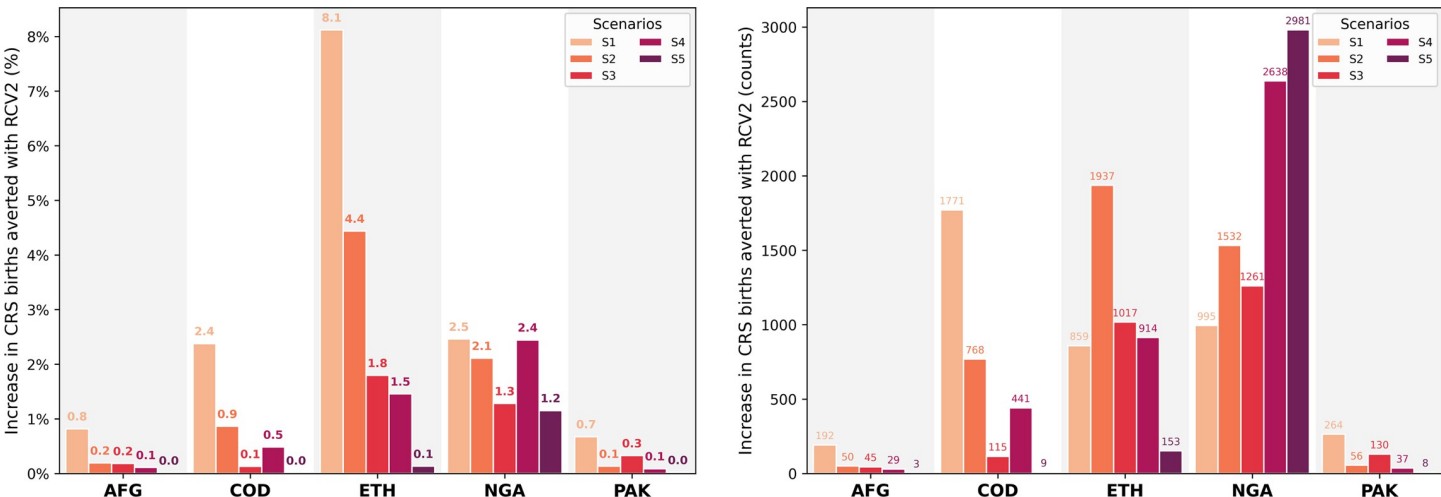

**Fig 2. Impact of including a second dose of RCV in the routine immunization schedule in CRS births averted.**

### 3.4 Sensitivity analysis

Fig 4 shows the median and several quantile intervals (50%, 75%, and 95%) for 100 projections of CRS incidence for each country using scenario S4. The curves show that incidence remains low for the first 15 years after introduction, after which different outcomes occur. Although initial results for both AFG and COD show a significant decrease in CRS incidence (Fig 1), the sensitivity analysis exhibits high variability after 20 years of introduction. The risk exists that CRS incidence is greater than the pre-vaccination conditions in COD, with the 95% quantile interval above 100 CRS births per 100,000 live births, while AFG remains below 45 CRS births per 100,000 live births. On the other hand, both ETH and NGA present high variability after 15 years of RCV introduction. The former presents a wide uncertainty interval with values ranging from close to zero CRS incidence up to 180 CRS births per 100,000 live births. PAK is the only country that maintains CRS incidence at relatively low levels with the current routine immunization coverage.

## 4. Discussion

Overall, the results show the potential impact RCV introduction can have in a country; CRS births can be significantly averted if RCV is introduced in sufficient quantities over time. Further, there is value in using an intervention that corresponds to the needs of a particular country setting. In countries where current immunization programs have relatively high routine coverage (PAK, AFG, and COD), the introduction of RCV may be straightforward. For example, introducing RCV in the routine schedule (scenario S1) of AFG and PAK results in a sustained decrease in CRS incidence throughout the entire study period. However, this same intervention would increase the number of CRS births in NGA and ETH compared to the zero-vaccination scenario (S0). Scenarios with sustained support of SIAs (S4 and S5) highlight the need for countries to ramp up their routine immunization coverage (AFG, COD, ETH, NGA) to avoid depending on supplemental immunization to reduce CRS incidence significantly.

The WHO recommends that at least 80% of routine immunization coverage of RCV1 is achieved for an effective introduction [1], which is reflected in our results for the minimum introduction rate. Our comparative analysis shows that achieving this coverage level ensures a

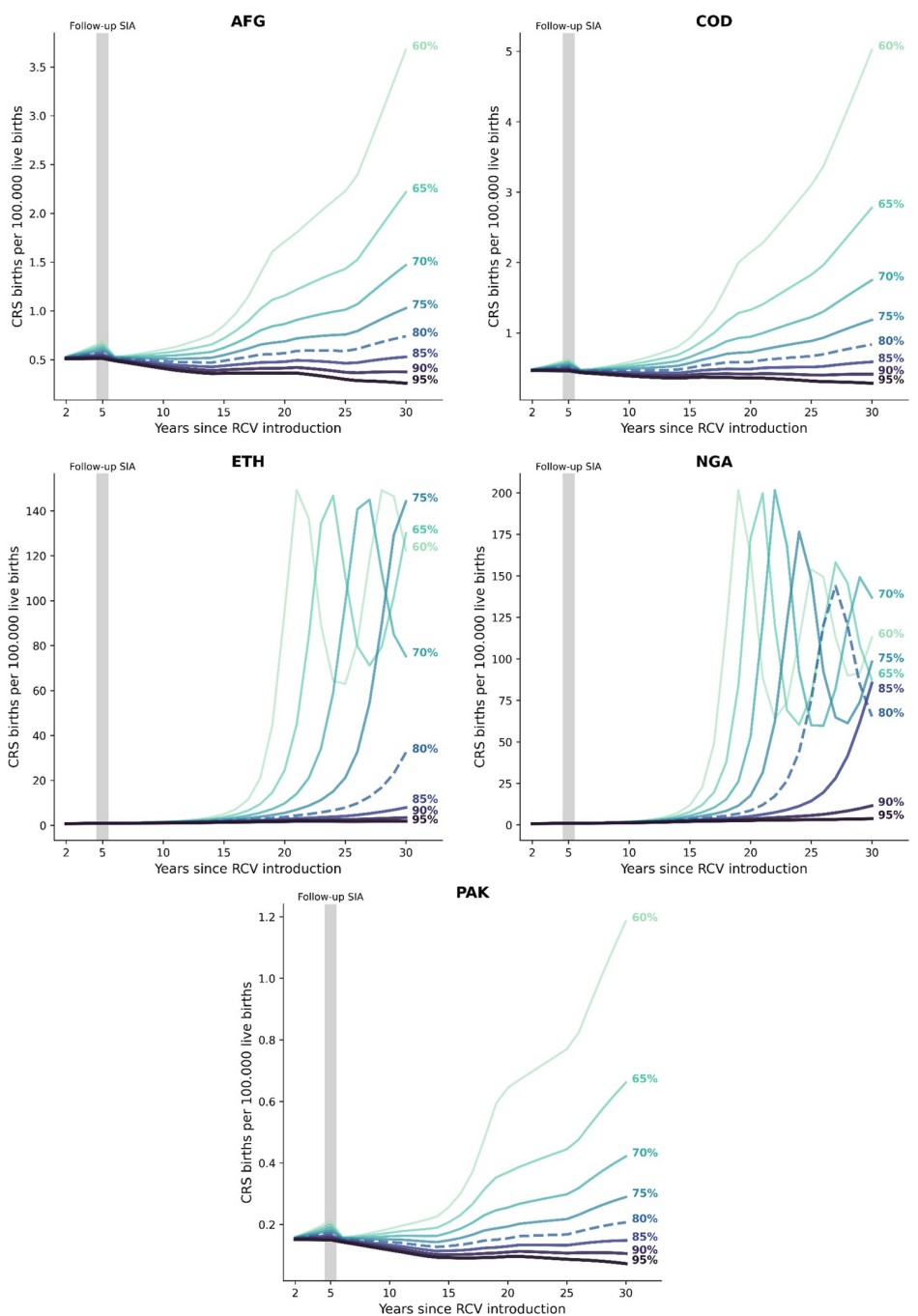

**Fig 3. CRS births per 100,000 live births under different RCV1 coverage rates.**

low incidence of CRS births (AFG, COD, PAK). However, countries with high-transmission profiles (NGA and PAK) present late outbreaks at the recommended immunization level, we estimate that coverage of over 90% would be necessary to achieve a low incidence of CRS births in these countries. However, we acknowledge that the model is limited by the assumption that the underlying average force of infection remains unchanged for the study period and, hence, does not consider changes in it caused by changes in the proportion of people

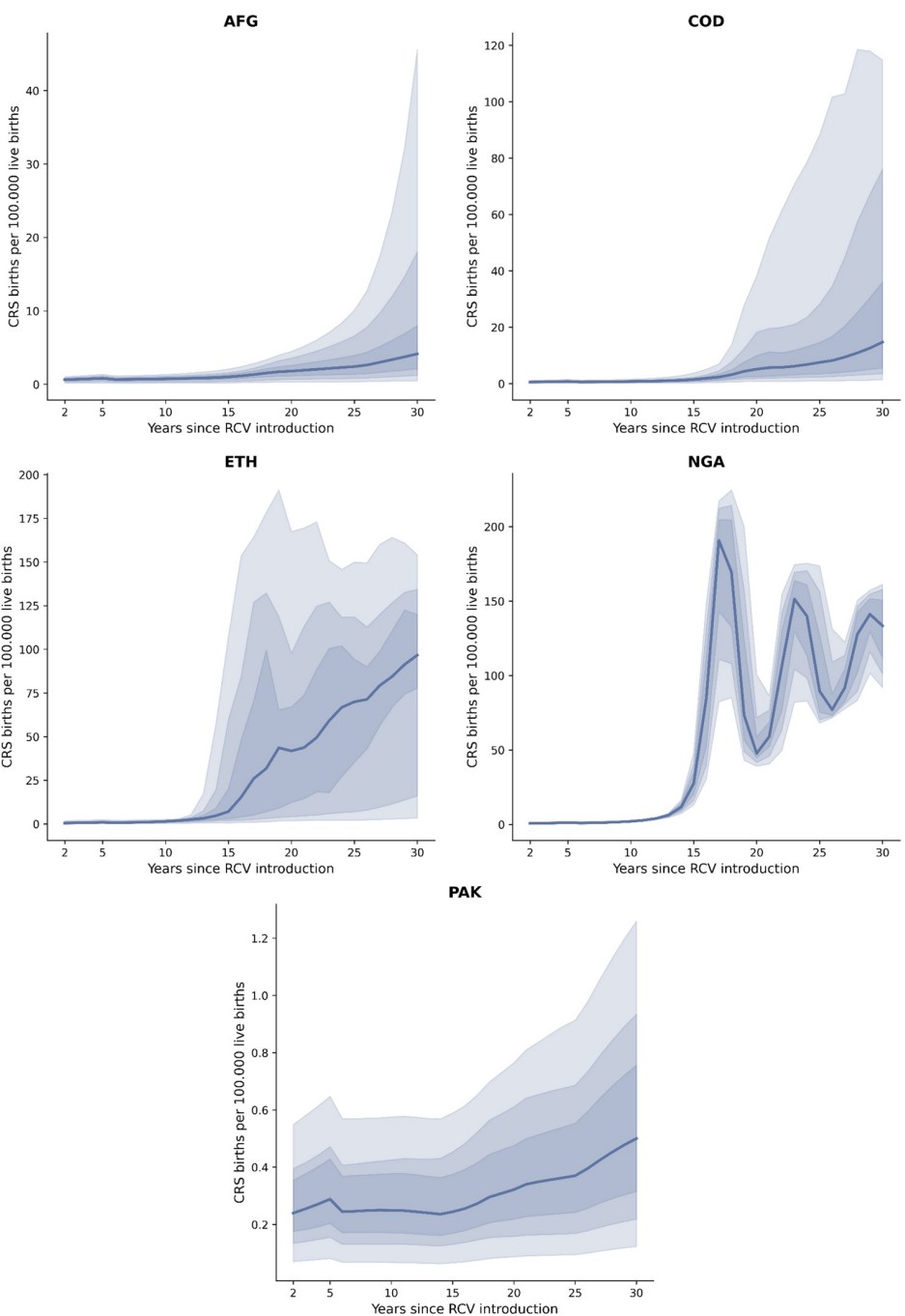

**Fig 4. 50%, 75%, and 95% quantile intervals for CRS incidence based on 100 samples of the average force of infection for scenario S4.**

vaccinated. Therefore, our results present a conservative estimate where infectivity remains high despite immunization. The estimated potential impact supports the validity of WHO recommendations and highlights the need for countries to increase RCV1 coverage to at least 80%.

Significant benefits can be achieved with a second dose of RCV despite the high efficacy of RCV1 and the fact that RCV2 effectively targets children who already received a first dose but

did not acquire immunization. The potential impact will depend on the country's current routine coverage. Between 900 (S1) and 3000 (S5) CRS births can be averted due to RCV2 in NGA, corresponding to between 1.2% and 2.5% reduction in CRS births compared to only giving RCV1. Similarly, a 2.4% and 8.1% reduction in CRS births can be achieved with RCV2 (S1) in COD and ETH, respectively. The potential impact of RCV2 in countries with relatively high RCV1 coverage (AFG and PAK) is less than a 1% reduction in CRS cases. Although RCV2 is not recommended by WHO, including it in the EPI schedule might be a beneficial complementary action to prevent CRS cases while the country increases RCV1 coverage. Results highlight the need to support actions toward strengthening countries' routine immunization programs, which requires enormous efforts from both local agencies and supporting organizations.

Additionally, sensitivity analysis suggests that the projected immunization level in scenario S4 is insufficient to prevent future outbreaks, which could lead to increased CRS births. The high transmission estimates for the population's young and old portions and the below-recommended routine coverage might explain these late outbreaks. However, introducing RCV can significantly reduce CRS incidence during the first 15 years. Maintaining this reduction after this will depend on the increase in routine coverage.

The pre-vaccination CRS incidence estimates used in this study align with those published for the Democratic Republic of Congo [32]. However, estimates for Ethiopia differ from those reported in [8] mainly because we derived national estimates for the force of infection based on those reported in [17]. To the best of our knowledge, for Afghanistan, Pakistan, and Nigeria, the latest pre-vaccination CRS incidence estimates are those provided by [17,18] (see Table B in S1 Appendix). This highlights the need for seroprevalence data to estimate better the CRS incidence in countries where RCV has not been introduced. A more accurate estimate of the burden of the disease in countries with high transmission sero-profiles, which are most likely to experience future outbreaks, as highlighted in the sensitivity analysis, will help to quantify the potential impact of immunization campaigns.

There are several limitations to our work. First, the results are based on a compartmental model. While this model is widely accepted [12–14], it is deterministic and does not incorporate all population heterogeneity. Additionally, the model simplifies vaccination timing and administration to ensure the number of doses corresponds to vaccination coverage. This assumption could impact the transmission dynamics of the disease by underestimating the proportion of susceptible within target vaccination groups. Second, projections depend on the assumption that the estimated average force of infection remains constant; therefore, the model does not incorporate all uncertainty related to changes in transmission dynamics across multiple decades. The estimated CRS incidence might be overestimated if the underlying transmission dynamics decrease when the proportion of people vaccinated increases. Also, it is essential to note that the results are dependent on the estimates of the force of infection, which are unknown. Third, we do not account for sub-national infection dynamics, which could differ significantly, especially in large countries with drastic differences between urban-rural settings. Lastly, each country's projected routine immunization levels rely on reported MCV coverage. Despite recent improvements in global immunization coverage reporting, there are still quality issues in the data from the studied countries [33], which could impact our results.

## 5. Conclusions

In this paper, we utilized a previously proposed SEIR model to analyze the potential impact of various RCV introduction scenarios on the CRS birth incidence for five countries. Our results show that the introduction of rubella vaccine with appropriate interventions and strategies can

significantly reduce CRS births. Moreover, a sensitivity analysis shows that high-transmission countries have a high risk of suffering outbreaks if current immunization levels are sustained. Countries can avoid this by increasing routine coverage and conducting recommended SIAs for a successful introduction. An extension to our work would be to develop a cost-effectiveness analysis to assess these different introduction policies and the use of combination vaccines in the studied countries. Our approach shows that quantification of intervention strategies can be performed and provides insight into appropriate implementation strategies.

## Supporting information

**S1 Checklist. Inclusivity in global research questionnaire.**
(DOCX)

**S1 Appendix. Compartmental SEIR model.**
(DOCX)

**S2 Appendix. Potential impact of the frequency of follow-up SIAs in high-transmission countries.**
(DOCX)

**S3 Appendix. Projected outcomes and rubella transmission dynamics for each scenario.**
(DOCX)

## Author Contributions

**Conceptualization:** Sebastian A. Rodriguez-Cartes, Maria E. Mayorga, Julie L. Swann.

**Data curation:** Yiwei Zhang.

**Formal analysis:** Sebastian A. Rodriguez-Cartes.

**Funding acquisition:** Maria E. Mayorga, Julie L. Swann, Benjamin T. Allaire.

**Methodology:** Sebastian A. Rodriguez-Cartes.

**Project administration:** Benjamin T. Allaire.

**Software:** Sebastian A. Rodriguez-Cartes.

**Supervision:** Maria E. Mayorga, Julie L. Swann.

**Validation:** Maria E. Mayorga, Julie L. Swann, Benjamin T. Allaire.

**Visualization:** Yiwei Zhang.

**Writing – original draft:** Sebastian A. Rodriguez-Cartes, Yiwei Zhang.

**Writing – review & editing:** Maria E. Mayorga, Julie L. Swann, Benjamin T. Allaire.

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
