## [Decision Letter · Decision Letter 0]

20 Sep 2023

PGPH-D-23-01509

Evaluating the impact of rubella-containing vaccine introduction in congenital rubella syndrome for Afghanistan, Dem. Republic of Congo, Ethiopia, Nigeria, and Pakistan

Dear Professor Mayorga,

Thank you for submitting your manuscript to PLOS Global Public Health. After careful consideration, we feel that it has merit but does not fully meet PLOS Global Public Health’s publication criteria as it currently stands. Therefore, we invite you to submit a revised version of the manuscript that addresses the points raised during the review process. Please review the detailed reviews here.

We look forward to receiving your revised manuscript.

Kind regards,

Megan Coffee, MD, PhD

Academic Editor

Journal Requirements:

Additional Editor Comments (if provided):

Reviewers' comments:

Reviewer's Responses to Questions

**Comments to the Author**

1. Does this manuscript meet PLOS Global Public Health’s publication criteria? Is the manuscript technically sound, and do the data support the conclusions? The manuscript must describe methodologically and ethically rigorous research with conclusions that are appropriately drawn based on the data presented.

Reviewer #1: Yes

Reviewer #2: Yes

Reviewer #3: Yes

2. Has the statistical analysis been performed appropriately and rigorously?

Reviewer #1: Yes

Reviewer #2: Yes

Reviewer #3: Yes

3. Have the authors made all data underlying the findings in their manuscript fully available (please refer to the Data Availability Statement at the start of the manuscript PDF file)?

Reviewer #1: Yes

Reviewer #2: Yes

Reviewer #3: Yes

4. Is the manuscript presented in an intelligible fashion and written in standard English?

Reviewer #1: Yes

Reviewer #2: Yes

Reviewer #3: Yes

5. Review Comments to the Author

Reviewer #1: This is an interesting paper addresses rubella vaccination and its impact on CRS. Here are a few comments that could improve on the manuscript:

- I think the title can be improved. It currently reads "Evaluating the impact of rubella-containing vaccine introduction in congenital rubella syndrome for Afghanistan, Dem. Republic of Congo, Ethiopia, Nigeria, and Pakistan" but should be modified to read "Evaluating the impact of rubella-containing vaccine introduction on congenital rubella syndrome in Afghanistan, Dem. Republic of Congo, Ethiopia, Nigeria, and Pakistan"

-There are minor language errors throughout the manuscript which a copy editor can address.

-In the methods section, you assumed that the first year of introduction the RCV coverage would correspond to 50% of the MCV immunization coverage and kept constant at 100% for the following years of the planning. I find this surprising because usually there is a lot of risk communication and community engagement prior to introduction of new vaccines which is likely to increase the coverage levels for the measles and rubella combination vaccine. Please provide an explanation for the assumption that RCV coverage will be lower than the figures for MCV in the first year of introduction.

- In lines 241-243 of the results, you write " Countries with high transmission levels (NGA and ETH) require a 95% introduction rate to ensure sustained reduction in CRS incidence, with routine coverage below 80% resulting in late outbreaks." It is surprising that a coverage figure of 95% is required to obtain a sustained reduction in CRS incidence. Rubella is not as transmissible as measles so it would be helpful to further justify this finding.

-In the results section, I find it difficult to distinguish the five scenarios in figure 1. In addition, the figures look blurred and the writings are not clearly visible.

-In lines 284-285, you write "A second dose of RCV is required, especially in those countries where current first-dose coverage is below the recommended target of 80%". I think this needs a bit more explanation given that the second dose of RCV is likely to be administered to the same group of children who received the first dose and not those who missed the first dose.

-It would be helpful to discuss how changes in population immunity following vaccine introduction affect the projections of CRS incidence over time. Do the estimates of force of infection remain constant over the entire simulation period irrespective of the vaccination coverage achieved?

- I think an additional limitation is the fact that sub-national variation in rubella epidemiology are not accounted for.

Reviewer #2: The investigators report a mathematical modeling study of the estimated potential impact of introduction and use of RCV on the occurrence of CRS over 30 years in five countries currently not vaccinating against rubella. They used a catalytic compartment differential equation-based model with states, M, S, E, I, and R. They assumed maternal antibody, which would prevent immunization by killing the RCV virus, is gone by 6 months of infant age. They assumed a 2-dose RCV routine schedule following introduction, with the first dose after 6 months, timed to coincide with MCV (as a combination vaccine). Country-specific RCV coverage levels were capped at MCV coverage levels reported to WHO/UNICEF (or 30% if MCV2 has not been introduced), with ramp-up from 0% to 100% of MCV coverage after two years. Total effective vaccination was assumed to be the product of the vaccine efficacy (0.95, the seroconversion rate) and the maximum between routine coverage and an assumed SIA coverage of 80%. They modeled five vaccination scenarios that were combinations of routine vaccination and SIAs (a wide-age-ranging SIA among 0-14-year-olds and follow-up SIAs among 0-4-year-olds) and one scenario of no RCV introduction, under assumptions of three different birth rates. Model outcomes were CRS cases and DALYs. They modeled various coverage levels to determine a coverage level in which CRS decreases. They conducted sensitivity analyses of the FoI from their catalytic model. They found that with the exception of one country under the scenario of RI-only (in which CRS increases due to RCV introduction), introduction of RCV decreases CRS substantially, varying by scenario. Three countries with high MCV coverage may only need SIAs around RCV introduction time; two other countries need SIAs for an indeterminate period. Sensitivity analysis showed that RI and SIA coverage needs to be improved beyond current levels to prevent future CRS outbreaks. They conclude that (proper) RCV introduction reduces CRS, that a second RCV dose is necessary in countries with 1-dose coverage is less than 80%, and that RI coverage needs to be increased.

The topic is of obvious importance as all WHO regions have a goal to eliminate rubella (and therefore prevent CRS). Mathematical modeling is the most common way of predicting vaccine impact on rubella and CRS. Their use of a catalytic compartment model is appropriate. Their scenarios make programmatic sense. Use of current MCV coverage data is appropriate although conservative. The manuscript is clearly written. Between the manuscript and the supplementary file, their model is described quite well. Their conclusions are based on model outputs.

I have a few suggestions to improve this quite good manuscript.

The title, “Evaluating the impact of rubella-containing vaccine introduction in congenital rubella syndrome for Afghanistan, Dem. Republic of Congo, Ethiopia, Nigeria, and Pakistan,” makes it appear that this is a study of the actual impact of RCV introduction. However, it is a study of the potential impact of RCV introduction based on a mathematical model. STOBE guidelines suggest putting the study design in the title. I think that their title should be “Evaluating the potential impact of … : a mathematical modeling study.”

Throughout the manuscript, when appropriate, the authors should make sure that it is clear that impact refers to potential impact. For example, the first sentence in the Abstract, “we assessed the impact of introducing rubella-containing vaccine” should be changed to indicate to the reader that it is potential impact.

The scenario in which RCV is never introduced is used to estimate the reduction of CRS births in the vaccinating scenarios. It would help the reader to put the CRS reduction values in context if the number of CRS births in the non-vaccinating scenario was included in the results – in the narrative and especially in Table 2 of the manuscript. Reporting the base case number of CRS cases would help the reader evaluate the face validity of the model. Additionally, when appropriate, a percentage reduction of CRS (in addition to a numeric reduction) would be a meaningful result for the reader to interpret.

On Line 133, “endemic equilibrium” is mentioned as a starting condition for the study. However, endemic equilibrium is not defined. Rubella occurs in periodic epidemics, which are likely to occur in different times in the five countries. The authors should define what they mean by endemic equilibrium.

The authors used two doses in RI for all scenarios other than the non-vaccination scenario. Since the take rate of a single dose of RCV is >95% and believed with some evidence to be lifelong, a second dose only immunizes the few who didn’t seroconvert from one dose. It would make more policy sense to put the effort into giving a first dose to more children than revaccinating the immune. Such an effort is consistent with the WHO Immunization Agenda 2030 strategy of having no zero-dose children. The problem addressed in the countries requiring a second dose is low coverage (lines 225-233). The authors recommend continued SIAs, but a far better strategy would be to strengthen RI to reach more children, reducing zero-dose children and achieving immunization benefits from more than just rubella vaccine. The SIAs, presumably, are to reach the unvaccinated children, but as the authors point out (lines 128-132), most children vaccinated in SIAs are already immune through RI. Since WHO, through IA2030, is promoting RI to reduce zero-dose children, it would be good to have at least one scenario in which RI improves.

The authors conclusion that two RCV doses is required for certain countries goes against the WHO recommendation of needing only a single dose. Therefore, further explanation of why two doses are required would help the reader.

The authors determination that effective vaccination was the product of the one-dose take rate and the maximum between RI coverage and SIA coverage seems a bit artificial (lines 129-131). There is an assumption in their approximation of which children are vaccinated in follow-up SIAs. They authors should indicate the thinking behind their method to estimate effective vaccination.

Reviewer #3: Reviewer’s comments

This modeling study examined the impact of introducing a rubella-containing vaccine (RCV) on congenital rubella syndrome (CRS) incidence in Afghanistan (AFG), Democratic Republic of Congo (COD), Ethiopia (ETH), Nigeria (NGA), and Pakistan (PAK). It is a relevant study that could potentially guide the vaccination strategies that could be deployed in different endemic countries yet to introduce an RCV-containing vaccine given different underlying conditions and assumptions. It is well-written but still requires minor editing. The limitations were documented but not exhaustive.

Specific comments:

1. Kindly revisit this sentence: “As of 2021, all these countries have MCV1 (1st dose of measles-containing vaccine) as part of their routine vaccination schedule, and only the Democratic Republic of Congo is yet to report introduction coverage of MCV2”. Seems there is no congruency.

2. You may want to provide additional potential gains from the study (not limited to this): “Results can be used to better inform resource allocation decisions toward 82 the elimination of rubella”

3. We also assumed that SIA campaigns are conducted at the beginning of each year. We also adapted the model to introduce 10 cases per year to incorporate the possibility of imported cases. What guided these assumptions- may be well-contested?

4. Additionally, we assumed that during the first year of introduction the RCV coverage would correspond to 50% of the MCV immunization coverage and kept constant at 100% for the following years of the planning horizon for each country. This is overly ambitious and unrealistic, given the immunization trends/history in these countries of interest (NGA, PAK, COD etc.).

5. We also estimated the number of CRS births by considering the three varying levels of projected 159 births (low (L), medium (M) and high (H)) from (27)? Guess there is a typo here.

6. In the method section, what guided the assumptions made on the set points for the follow-up in the various scenarios, particularly scenarios 3, 4 and 5. For completeness, the author may want to state this.

7. Regarding the limitation, could the validity and reliability (quality) of the underlying routine immunization data impact the outcome of the modeling study for the selected countries?

6. PLOS authors have the option to publish the peer review history of their article (what does this mean?). If published, this will include your full peer review and any attached files.

**Do you want your identity to be public for this peer review?** For information about this choice, including consent withdrawal, please see our Privacy Policy.

Reviewer #1: **Yes: **Nkengafac Villyen Motaze

Reviewer #2: No

Reviewer #3: No

---

## [Decision Letter · Decision Letter 1]

1 Dec 2023

Evaluating the impact of rubella-containing vaccine introduction on congenital rubella syndrome in Afghanistan, Dem. Republic of Congo, Ethiopia, Nigeria, and Pakistan: a mathematical modeling study

PGPH-D-23-01509R1

Dear Professor Mayorga

We are pleased to inform you that your manuscript 'Evaluating the impact of rubella-containing vaccine introduction on congenital rubella syndrome in Afghanistan, Dem. Republic of Congo, Ethiopia, Nigeria, and Pakistan: a mathematical modeling study' has been provisionally accepted for publication in PLOS Global Public Health.

Best regards,

Megan Coffee, MD, PhD

Academic Editor

Reviewer Comments (if any, and for reference):

Reviewer's Responses to Questions

**Comments to the Author**

1. If the authors have adequately addressed your comments raised in a previous round of review and you feel that this manuscript is now acceptable for publication, you may indicate that here to bypass the “Comments to the Author” section, enter your conflict of interest statement in the “Confidential to Editor” section, and submit your "Accept" recommendation.

Reviewer #1: All comments have been addressed

Reviewer #2: All comments have been addressed

Reviewer #3: All comments have been addressed

2. Does this manuscript meet PLOS Global Public Health’s publication criteria? Is the manuscript technically sound, and do the data support the conclusions? The manuscript must describe methodologically and ethically rigorous research with conclusions that are appropriately drawn based on the data presented.

Reviewer #1: Yes

Reviewer #2: Yes

Reviewer #3: Yes

3. Has the statistical analysis been performed appropriately and rigorously?

Reviewer #1: Yes

Reviewer #2: Yes

Reviewer #3: Yes

4. Have the authors made all data underlying the findings in their manuscript fully available (please refer to the Data Availability Statement at the start of the manuscript PDF file)?

Reviewer #1: Yes

Reviewer #2: Yes

Reviewer #3: Yes

5. Is the manuscript presented in an intelligible fashion and written in standard English?

Reviewer #1: Yes

Reviewer #2: Yes

Reviewer #3: Yes

6. Review Comments to the Author

Reviewer #1: Dear Authors,

Thank you for revising the manuscript following the comments and for justifying where you did not agree with the feedback.

I think the article is suitable for publication in its present form.

Regards

Reviewer #2: I was one of the reviewers of the original version of the manuscript. The authors did a nice job addressing my comments and suggestions. I have no additional comments or suggestions.

Reviewer #3: Comments by Reviewer 3: At the current state, the manuscript is suitable for publication. Congratulations to the author for this important contribution to knowledge and potentially, field implementation. However, I suggest these minor edits:

1. The word “potential” should be included (before “impact”) in the title of the manuscript as suggested by reviewers. The recommended change has been reflected elsewhere in the manuscript.

2. See line 72: “Underdeveloped countries”, this should be changed to “developing countries”.

7. PLOS authors have the option to publish the peer review history of their article (what does this mean?). If published, this will include your full peer review and any attached files.

**Do you want your identity to be public for this peer review?** For information about this choice, including consent withdrawal, please see our Privacy Policy.

Reviewer #1: **Yes: **Nkengafac Villyen Motaze

Reviewer #2: No

Reviewer #3: **Yes: **Dr Semeeh Akinwale OMOLEKE
